# A hidden gene in astroviruses encodes a viroporin

Valeria Lulla [1✉] & Andrew E. Firth [1✉]

Human astroviruses are small non-enveloped viruses with positive-sense single-stranded RNA genomes. Astroviruses cause acute gastroenteritis in children worldwide and have been associated with encephalitis and meningitis in immunocompromised individuals. It is still unknown how astrovirus particles exit infected cells following replication. Through comparative genomic analysis and ribosome profiling we here identify and confirm the expression of a conserved alternative-frame ORF, encoding the protein XP. XP-knockout astroviruses are attenuated and pseudo-revert on passaging. Further investigation into the function of XP revealed plasma and trans Golgi network membrane-associated roles in virus assembly and/ or release through a viroporin-like activity. XP-knockout replicons have only a minor replication defect, demonstrating the role of XP at late stages of infection. The discovery of XP advances our knowledge of these important human viruses and opens an additional direction of research into their life cycle and pathogenesis.

[1] Division of Virology, Department of Pathology, Addenbrooke's Hospital, University of Cambridge, Cambridge, UK. ✉email: vl284@cam.ac.uk; aef24@cam.ac.uk

Humans astroviruses (HAstVs) belong to genus *Mamastrovirus* within the family *Astroviridae*, and are a very common cause of gastroenteritis in infants[1]. Recently, and especially in immunocompromised individuals, astroviruses have also been associated with extra-intestinal infections including fatal meningitis and encephalitis[2]. Despite their undoubted importance, astroviruses still represent one of the least studied groups of human positive-sense RNA viruses. The genome contains three main open reading frames: ORF1a encoding a nonstructural polyprotein (nsP1a), ORF1b encoding the RNA-dependent RNA polymerase (RdRp), and ORF2 encoding the capsid protein (CP)[3]. ORF1a and ORF1b are translated from the genomic RNA (gRNA), with expression of ORF1b depending on programmed ribosomal frameshifting whereby a proportion of ribosomes translating ORF1a make a −1 nucleotide shift into ORF1b. Frameshifting occurs at a conserved A_AAA_AAC sequence within the ORF1a/ORF1b overlap region and depends on a 3′-adjacent stimulatory RNA stem-loop structure[4]. ORF2 is translated from a subgenomic RNA (sgRNA) that is produced during virus infection.

Previously, using comparative genomics we identified a conserved fourth ORF (ORFX) in genogroup I astroviruses, which appears to be subject to purifying selection and therefore is likely to encode a functional protein product, termed the X protein (XP, 12 kDa, 112 aa)[5]. ORFX overlaps the 5′ region of the capsid-encoding ORF2 and is predicted to be translatable via ribosomal leaky scanning, which is expected to be enhanced by the very short leader on the sgRNA[3,6,7]. However, despite the comparative genomic predictions, the existence, relevance, and function of XP have never been demonstrated. Here we perform a functional molecular dissection of ORFX, providing insight into the simultaneous orchestration of several overlapping functional elements in a small human-pathogenic RNA virus. We show that ORFX is translated during virus infection, XP knockout mutant viruses are highly attenuated, and XP promotes efficient virus assembly and/or release likely via its viroporin-like activity. Through an extended bioinformatic analysis, we predict that related XP proteins are widely encoded in other mammalian astroviruses including the divergent MLB group of human-infecting astroviruses.

## Results

**Comparative genomic analysis reveals a potential ORFX.** Since there are many more astrovirus sequences available now than in 2010, we began by repeating our previous comparative genomic analysis but this time also extending to other astrovirus genogroups. All mammalian astrovirus complete or nearly complete genome sequences were obtained from the National Center for Biotechnology Information (NCBI), ORF1b (RdRp) amino acid sequences were extracted and aligned, and a phylogenetic tree constructed (Supplementary Fig. 1). We follow the genogroups defined in Yokoyama et al.[8]. Although the more-divergent CP sequences may provide a less robust phylogenetic analysis, we also constructed a CP-based phylogenetic tree to test for possible recombination (Supplementary Fig. 2). At the level of genogroup the ORF1b and CP trees were consistent, although within genogroups there were some differences between the topologies of the two trees. For simplicity, we used the ORF1b tree to define astrovirus clades for full-genome analyses.

We grouped astrovirus sequences into clades (Fig. 1a), generated codon-based alignments of concatenated ORF1a-ORF1b-ORF2 coding sequences within each clade, and analyzed the alignments with SYNPLOT2, which tests for regions where synonymous substitutions occur less often than average for the sequence alignment[9]. Regions with significantly enhanced synonymous site conservation typically harbor overlapping functional elements, which constrain sequence evolution[9]. The analysis of subgroup Ia astroviruses (Fig. 1a; classical HAstVs besides some feline and sea lion astroviruses) revealed conserved elements at the junction of ORF1a and ORF1b (representing the ribosomal frameshifting signal and ORF1a/ORF1b overlap), upstream of ORF2 (presumed to represent elements involved in sgRNA synthesis), toward the 3′ end of ORF2 (possible replication elements), and overlapping the 5′ end of ORF2 (the putative overlapping ORFX) (Fig. 1b). We applied the same analysis to 13 other astrovirus genogroup or sub-genogroup clades, revealing conserved regions overlapping the 5′ end of ORF2 in genogroup I, III, and IV astroviruses but, generally (see below for exceptions), not in genogroup II or VI astroviruses (Supplementary Fig. 3).

We repeated the synonymous site conservation analysis using clades, sequence alignments, and phylogenetic trees based on only ORF2 in order to rule out potential artefacts as a result of possible recombination between the nonstructural and structural modules of the virus genome, and to include additional part-genome sequences (total 415 sequences with ORF2 coverage; Fig. 1c and Supplementary Fig. 4). This analysis revealed conserved regions overlapping the 5′ end of ORF2 throughout genogroup I, III, and IV astroviruses but, generally, not in genogroup II or VI astroviruses (Supplementary Fig. 5). Those clades with synonymous site conservation also contain a conserved overlapping +1 frame ORF, whereas those sequences without synonymous site conservation generally also lacked a conserved overlapping +1 frame ORF (Supplementary Fig. 5). Interestingly, we also detected synonymous site conservation in two subgroups of genogroup II astroviruses (Supplementary Fig. 3g, h). In subgroup IId, the conservation coincided with a conserved overlapping +1 frame ORF (Supplementary Fig. 5e); however in subgroup IIc, there was no conserved ORF in the +1 reading frame. Instead, we observed an ORF in the −1 reading frame with no AUG codon (ORFY, Supplementary Fig. 5d) but with conserved signals for −1 programmed ribosomal frameshifting[10], namely a conserved A_AAA_AAZ (Z = A, C or U) slippery heptanucleotide and a 3′-adjacent RNA stem-loop or pseudoknot structure (Supplementary Fig. 6). It seems plausible that ORFX and ORFY in genogroup II astroviruses may have evolved independently of ORFX in other astrovirus genogroups (see Discussion).

Evidence for ORFX was also found in a clade of rodent astroviruses of unassigned genogroup (Supplementary Figs. 3d and 5l). Further, a number of unclustered divergent sequences also contain long +1 frame ORFs overlapping the 5′ end of ORF2 that potentially encode XP proteins (Supplementary Fig. 7). Note that genogroup V—represented by a single partial sequence, FJ890355 (bottlenose dolphin astrovirus 1)—is one of these.

The putative XP proteins encoded by genogroup I, III, and IV astroviruses typically range from 70–112 aa in length and 7.5–12.3 kDa in molecular mass (Fig. 1d–e; Supplementary Table 1). Similar to other overlapping genes, which generally evolve de novo[11], the XP peptide sequences show little to no homology to known protein domains. Many XP sequences contain a predicted transmembrane (TM) domain, and some other XPs contain a stretch of hydrophobic amino acids that resembles a TM domain despite being scored below threshold by Phobius (Supplementary Figs. 8 and 9).

**Ribosome profiling confirms translation of ORFX.** To test for XP expression in infected cells, we first raised various antibodies against XP peptides and we also tested the viability of tagged-XP viruses. However, neither antibody nor tagged virus approaches were successful due to poor immunogenicity of XP and non-

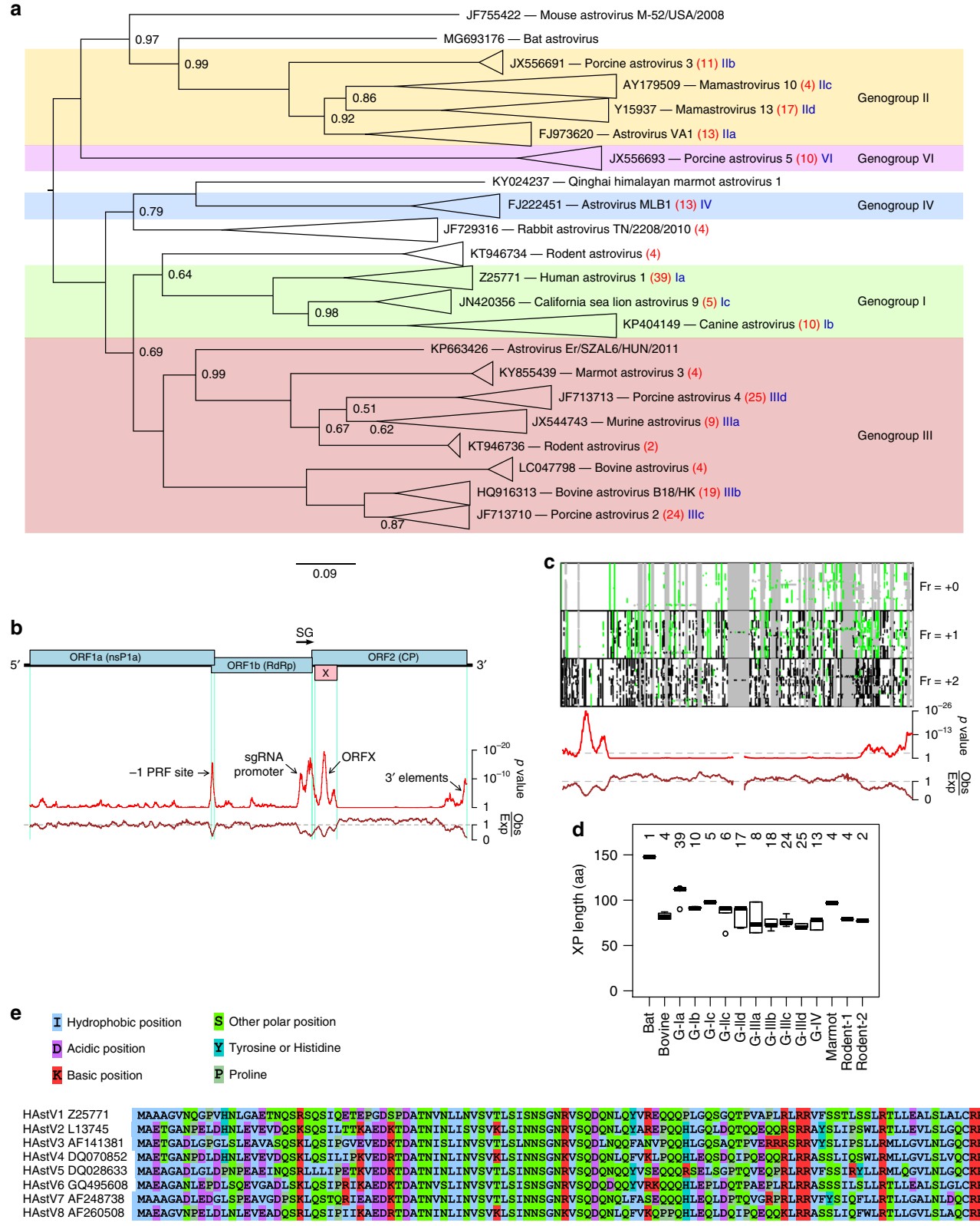

viability of the tagged viruses. Thus we turned to ribosome profiling (Ribo-Seq). Ribo-Seq is a high-throughput sequencing technique that globally maps the footprints of initiating or elongating 80S ribosomes on mRNAs[12,13]. We infected Caco2 (human epithelial colorectal adenocarcinoma) cells and performed Ribo-Seq at 12 h post infection (hpi). Ribo-Seq quality was assessed as previously described[14] (Supplementary Fig. 10).

Using flash-freezing with no drug pre-treatment (NT), we mapped the translational landscape of the HAstV1 genome (Fig. 2a). ORF2 is translated at ~9× the level of ORF1a, whereas ORF1b is translated at ~25% the level of ORF1a, indicating a ribosomal frameshifting efficiency of ~25%. The higher expression of ORF2 is likely a result of higher levels of sgRNA than gRNA in the translation pool. To identify translation initiation sites, we

**Fig. 1 Comparative genomic analysis of astroviruses. a** Phylogenetic tree of mammalian astroviruses. The tree, calculated with MrBayes, is based on ORF1b amino acid sequences obtained from 221 full-length genomes. Related groups of sequences (indicated by isosceles triangles) have been replaced in the figure by a single representative accession number and virus name; the total number of sequences in each group is shown in red (see Supplementary Fig. 1 for the complete tree). Genogroups (according to Yokoyama et al.[8]) are indicated by background shading. Subgroups of sequences, defined for the purposes of this study only, are indicated in blue (Ia, Ib, etc). The tree is midpoint rooted and nodes are labeled with posterior probability values if different from 1.00. **b** Map of the human astrovirus genome showing the three main ORFs (blue) and the overlapping ORFX (pink), and analysis of conservation at ORF1a-ORF1b-ORF2 synonymous sites. The red line shows the probability that the observed conservation could occur under a null model of neutral evolution at synonymous sites, whereas the brown line depicts the ratio of the observed number of substitutions to the number expected under the null model. Inferred elements corresponding to regions of enhanced synonymous site conservation are indicated. **c** Analysis of an alignment of 127 human and 5 feline astrovirus ORF2 sequences. The upper three panels show the positions of alignment gaps (gray), stop codons (black), and AUG codons (green) in each reading frame. Below, is shown the analysis of conservation at synonymous sites. **d** Box plots of XP length for different astrovirus clades: centre lines = medians; boxes = interquartile ranges; whiskers extend to most extreme data point within 1.5 × interquartile range from the box; circles = outliers; number of sequences, n, is shown above each box. **e** Alignment of XP sequences from representative HAstVs. Source data are provided as a Source Data file.

utilized the translation inhibitor lactimidomycin (LTM), which acts preferentially on the initiating ribosome but not on the elongating ribosome[15]. LTM binds to the 80S ribosome already assembled at the initiation codon and occupies the empty exit (E) site of initiating ribosomes, thus completely blocking translocation. Using this approach, we confirmed the two previously known initiation sites, i.e., for ORF1a and ORF2, and also identified substantial initiation at the ORFX start codon (Fig. 2b), this being the third largest peak in the LTM virus profile.

It is worth noting that the short leader of the sgRNA may result in protection by the ribosome of the 5′ end of ORF2 initiation footprints from the RNase I nuclease, so that many or most ORF2 initiation footprints might retain the viral VPg protein that is likely to be covalently linked to the 5′ end of sgRNAs[16]. Such reads will not ligate to the adapter oligonucleotides and thus will be excluded from sequencing. This may explain why the ORF2 initiation peak is smaller than the ORF1a initiation peak, even though ORF2 is expressed at much higher levels than ORF1a. Thus we cannot quantify the ORF2:ORFX expression ratio from the LTM data.

Ribosome profiling of eukaryotic systems typically has the characteristic that mappings of the 5′ end positions of ribosome protected fragments (RPFs) to coding sequences reflect the triplet periodicity (herein referred to as phasing) of genetic decoding. For our datasets, the great majority of RPF 5′ ends map to the first nucleotide of codons (Fig. 2c, left). Reads mapping to the ORF2/ORFX part of the genome (NT samples) were quantified in the three possible phases. In the region of ORF2 that is overlapped by ORFX we observed an increased number of reads mapping in the +1 phase relative to the ORF2 reading frame (Fig. 2c, middle) compared with the region of ORF2 that is not overlapped by ORFX (Fig. 2c, right) or the coding regions of host mRNAs (Fig. 2c, left). Quantification of the differences in phasing indicated that ORFX is translated at ~27% of the level of ORF2, although it should be noted that reporter assays (see below) are expected to provide more accurate quantification than RPF phasing analysis. To gauge statistical significance, we performed 100,000 bootstrap resamplings of codon positions within the overlapping and nonoverlapping regions of ORF2 and estimated XP-frame:CP-frame expression ratios for each (Fig. 2d). The ratio is statistically significantly higher in the overlap region (1-tailed Kolmogorov–Smirnov test; $p < 2.2 \times 10^{-16}$); indeed the 95% confidence intervals do not even overlap (Fig. 2d).

**Attenuated XP-knockout viruses pseudo-revert on passage.** To evaluate the significance of ORFX in the context of virus infection, a set of mutant virus genomes was created based on the pAVIC1 infectious clone[17] by introducing mutations that knock out ORFX without affecting the CP amino acid sequence

(Fig. 3a). Four independent mutations were introduced to guard against the possibility of affecting potential RNA secondary structures overlapping with this region, resulting in pAVIC1-AUGm (AUG to ACG), pAVIC1-PTC1 (stop codon after 20 amino acids), pAVIC1-2×PTC (double stop codon after 20 amino acids), and pAVIC1-PTC2 (stop codon after 73 amino acids) (Fig. 3b and Supplementary Fig. 11). Caco2 cells are susceptible to HAstV infection but are not easily transfectable. In contrast, BSR (a clone of baby hamster kidney cells, BHK) and Huh7.5.1 (human hepatocellular carcinoma) cells support the astrovirus replication cycle following transfection with T7 RNA transcripts, but do not support infection with virus particles[17,18] (possibly due to lack of appropriate receptors). Thus, BSR or Huh7.5.1 cells are used for virus rescue, and rescued virus is used to infect Caco2 cells. All four ORFX knockout viruses were strongly attenuated in Caco2 cells (Fig. 3c). After seven blind passages, two mutant viruses (pAVIC1-AUGm and pAVIC1-PTC1) demonstrated an ~1 log increase in virus titers, which was achieved via a pseudo-reversion (pAVIC1-AUGm) or 5- or 8-codon deletions (pAVIC1-PTC1) (Fig. 3c and Supplementary Fig. 12). All three mutations would result in restoration of XP expression, confirming its importance in virus growth.

**XP is essential for virus assembly and/or release.** ORFX overlaps ORF2, which encodes the structural polyprotein, and therefore expression of XP is more likely to be relevant in late stages of the virus replication cycle. To rule out the possibility of XP directly influencing viral RNA replication, we developed a replicon system comprising an intact astrovirus genome up to the end of ORFX, followed by a 2A-RLuc cassette fused in either the ORF2 or ORFX reading frame, followed by the last 624 nt of the virus genome and a poly-A tail (Fig. 4a). These replicons provide a direct measurement of translated product associated with activity of the subgenomic promoter. Two ORFX knockout mutations were copied in both versions of the replicon to evaluate the significance of XP for RNA replication (Fig. 4b). Experiments were performed in BSR and Huh7.5.1 cell lines. Replication was confirmed by a 900–2100 fold difference in relative luciferase activity between wt and an RdRp knockout mutant (GDD → GNN) at 9 and 12 h post transfection (Fig. 4c and Supplementary Fig. 13a). The mutations introduced to knock out ORFX had minor effect on luciferase activity when 2A-RLuc was fused in the ORF2 reading frame (Fig. 4c and Supplementary Fig. 13a). Consistent with our Ribo-Seq results (Caco2 cells; Fig. 2c–d), luciferase activity for the ORFX reading frame was 14% (BSR cells; Fig. 4c) and 21% (Huh7.5.1 cells; Supplementary Fig. 13a) of luciferase activity for the ORF2 reading frame. Thus, ORFX is efficiently translated but has only a minor (if any) direct effect on viral RNA replication.

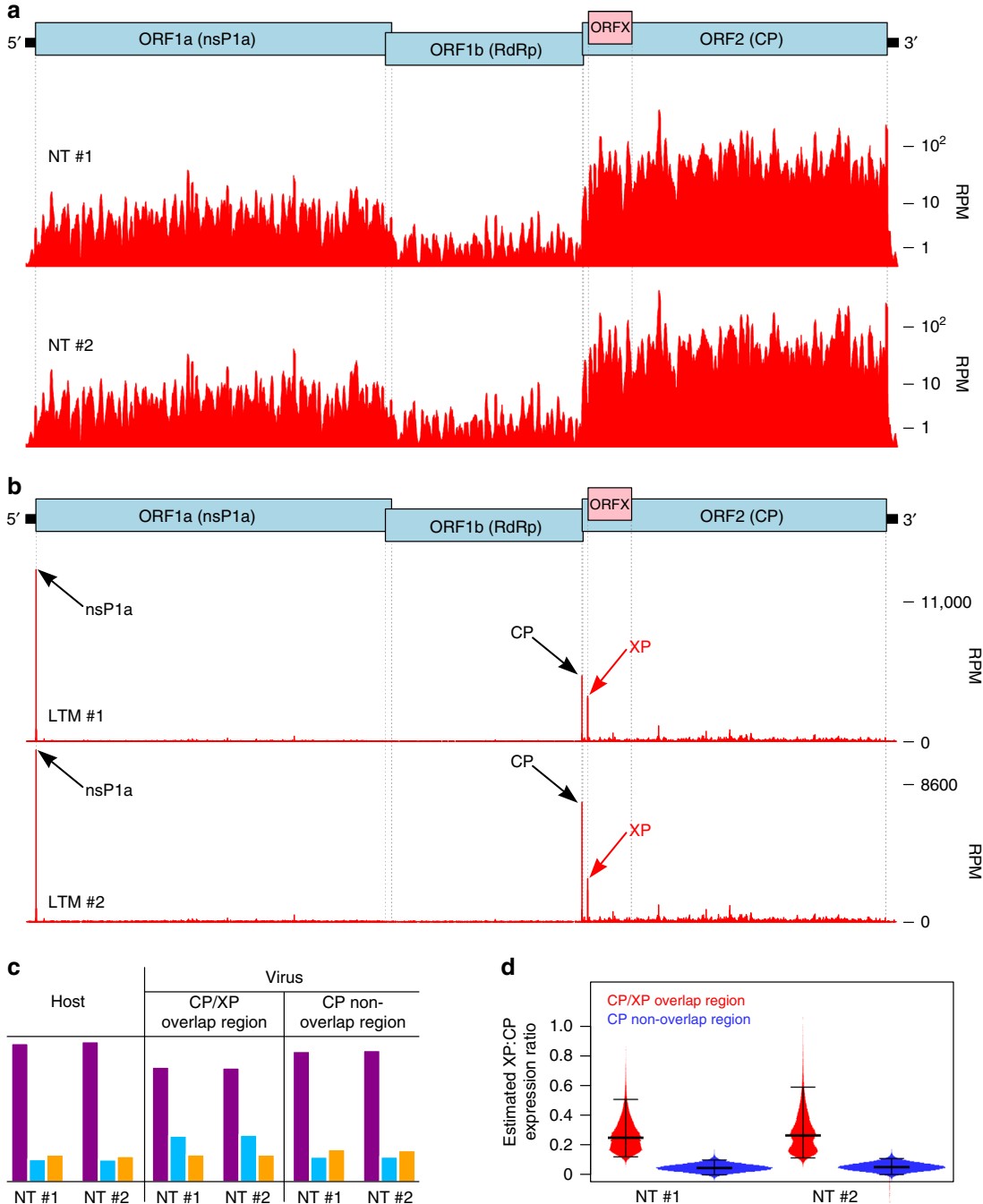

**Fig. 2 Ribosome profiling of astrovirus-infected cells.** Cells were harvested at 12 hpi and either flash frozen with no pre-treatment (NT), or pretreated with lactimidomycin for 30 min followed by flash-freezing (LTM). **a** RPF densities in reads per million mapped reads (RPM) for NT repeats, smoothed with a 15 nt sliding window. **b** RPF densities in RPM for LTM repeats, at single-nucleotide resolution. **c** For NT samples, phasing of 5′ ends of RPFs that map to host coding sequences, the part of ORF2 that is overlapped by ORFX, and the part of ORF2 that is not overlapped by ORFX. **d** Distributions of estimated XP-frame:CP-frame expression ratios for 100,000 bootstrap resamplings of codon positions within the overlapping and nonoverlapping regions of ORF2. Black lines show medians and 95% confidence intervals. Graphs show the results of two biologically independent experiments (#1 and #2). Source data are provided as a Source Data file.

As expected, both XP knockout mutations resulted in a substantial drop in ORFX-frame luciferase activity, and this drop was more pronounced for the PTC1 mutant than for the AUGm mutant (Fig. 4d and Supplementary Fig. 13b). ACG codons are known to permit initiation when in a strong initiation context (e.g., A at −3 and G at +4 as in the AUGm mutant)[19]. Since we

wished to only use mutations that were synonymous in ORF2, we were unable to mutate the ORFX AUG codon in any other way. When utilized as an initiation codon, ACG is expected to be decoded as methionine by initiator Met-tRNA. Therefore the AUGm mutant is still expected to produce wt XP, albeit at a greatly reduced level.

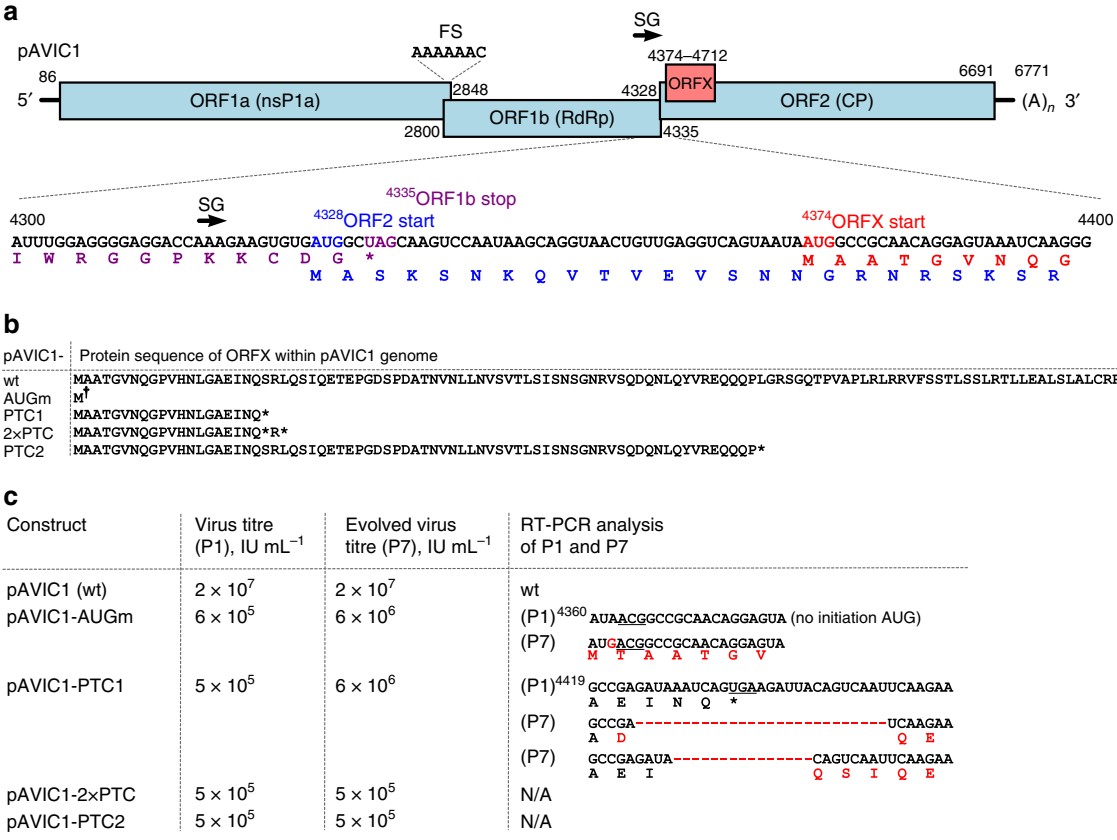

**Fig. 3 Design and properties of XP knockout viruses. a** Schematic representation of the astrovirus genome. Numbers correspond to astrovirus genome nucleotides in the pAVIC1 infectious clone; FS frameshift signal, SG subgenomic promoter. The sequence of nucleotides 4300–4400 is shown below, indicating the positions of overlapping elements and the corresponding translated proteins: ORF1b (purple), ORF2 (blue), and ORFX (red). **b** XP amino acid sequences for the wt and mutant pAVIC1-derived viruses († see main text for note on the AUGm mutant). See Supplementary Fig. 11 for nucleotide and CP amino acid sequences. **c** Titers (infectious units per mL, IU mL⁻¹) of recombinant viruses after RNA transfection of Huh7.5.1 cells followed by first passage (P1) and after seven passages (P7) in Caco2 cells. RT-PCR analysis of viral RNA isolated after P1 and P7 for selected recombinant viruses. See Supplementary Fig. 12 for sequencing chromatograms of individual passages.

The astrovirus subgenomic promoter is also situated within this region of the astrovirus genome (Fig. 4a). Although effective replication occurred even in the absence of XP translation (Fig. 4c), the introduced mutations could still have a potential effect on sgRNA production via alteration of RNA structure and/or other interactions. To map the minimal region required for wt levels of sgRNA production in the context of the pO2RL replicon, we gradually truncated the sequence between the start of ORF2 and ORFX. Surprisingly, we found that replicons containing only the first 4 or 28 nucleotides of ORF2 have very poor subgenomic reporter activity. In contrast, including the first 46 nucleotides of ORF2 restored subgenomic reporter activity to 92% of wt (Fig. 4f and Supplementary Fig. 13c–e). These data suggest that the astrovirus subgenomic promoter is substantially longer than previously reported[20] and extends into the 5′ part ORF2, but does not appear to extend into ORFX.

To determine which stage of the virus replication cycle is affected in XP knockout viruses, we quantified virus growth in BSR cells electroporated with in vitro transcribed full-length pAVIC1 T7 RNA. The abortive replication cycle in BSR cells (supporting a single round of virus growth and release, but not re-infection) provides an excellent platform to address this question. Both extracellular (released) and intracellular samples were used for evaluating virus titers, RNA levels, and protein synthesis at 48 h post electroporation (hpe) (Fig. 4g). In cell-derived samples, wt and mutant virus RNA and protein levels were similar, but virus titers were reduced in the mutants (393-

fold for the PTC1 mutant). In media-derived samples, mutant virus titers and RNA levels were substantially lower than for wt virus (612- and 37-fold, respectively, for the PTC1 mutant). When analyzed at 24 hpe, intracellular virus titers (Fig. 4h) and electroporation efficiencies (Fig. 4i) were similar for wt and mutant viruses. Although progression of cytopathic effect (CPE) differed between cells producing wt or mutant viruses (Supplementary Fig. 13f), cell viability did not differ greatly (Supplementary Fig. 13g). Finally, localization and distribution of CP in electroporated Huh7.5.1 cells (more suitable for imaging than BSR cells) was similar for wt and XP mutant viruses (Fig. 4j). Combined, these analyses indicate the involvement of XP late in the virus replication cycle, with XP potentially acting in either virus particle formation and/or virus release.

**XP multimerizes and localizes to TGN and plasma membranes.** To further investigate the function of XP, we studied its intracellular localization. To enable visualization of XP in transfected cells, we fused it either N- or C-terminally with mCherry or an HA tag in the context of a mammalian expression vector. The diffuse cytoplasmic localization of mCherry observed in live imaged Huh7.5.1 cells was drastically altered in both XP fusions, where it was relocalized to plasma and perinuclear membranes (Fig. 5a). A similar localization was also observed upon overexpression of HA-tagged XP followed by imaging in fixed and permeabilized HeLa cells (Fig. 5a, d). This strongly suggests the

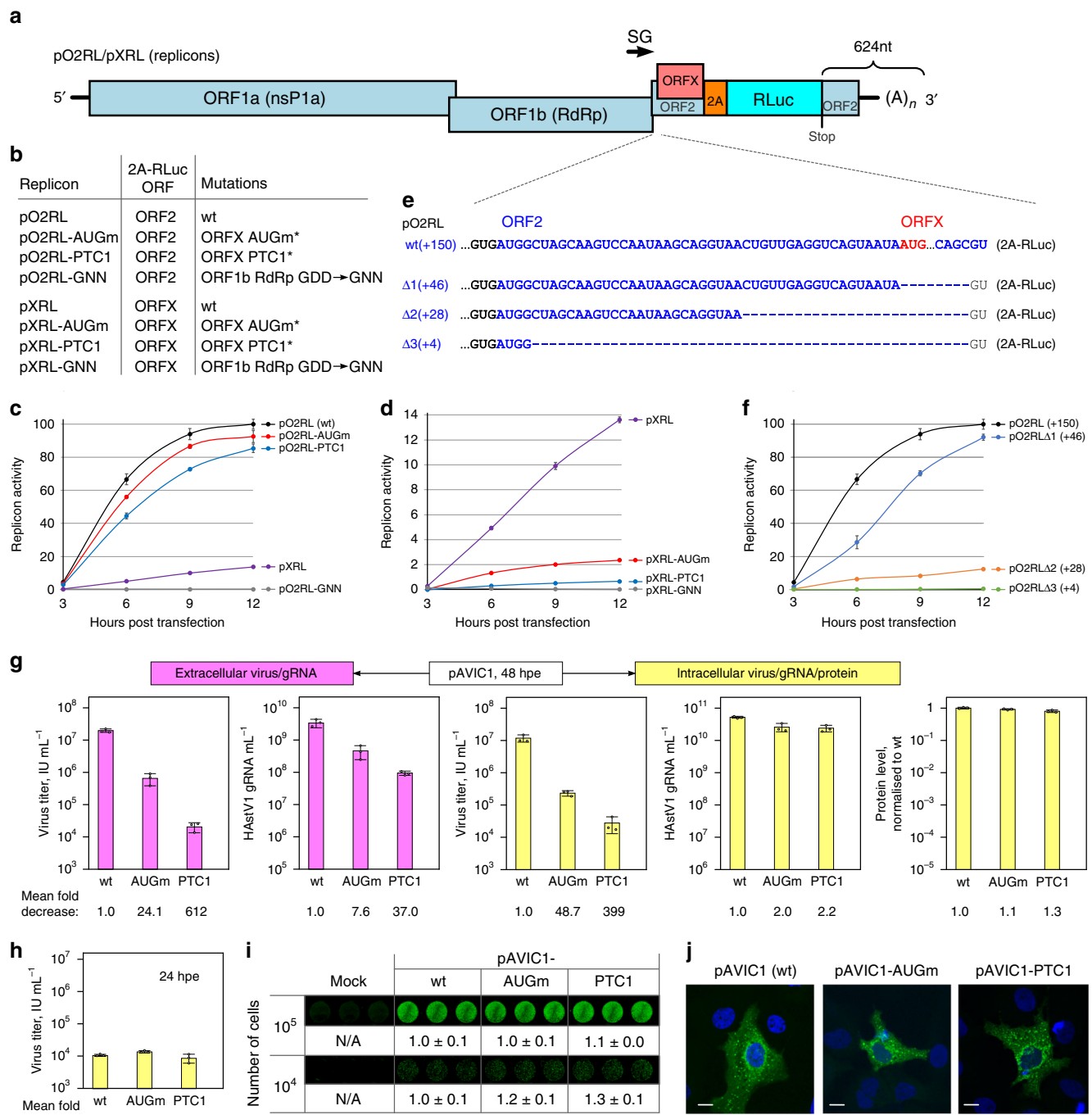

**Fig. 4 Analysis of replication stages affected by XP. a** Schematic of the astrovirus replicon. The 2A-RLuc cassette is fused in either the ORF2 (pO2RL) or ORFX (pXRL) reading frame. **b** List of replicon mutants showing the RLuc reading frame and introduced mutations. Asterisk—see Supplementary Fig. 11 for exact sequences. **c, d** Relative replicon luciferase activities measured after RNA transfection of BSR cells (see Supplementary Fig. 13a, b for Huh7.5.1 cells). Values are normalized so that the maximum ORF2-frame wt value is 100%. **e** Schematic of deletion mutants in the pO2RL replicon; ORF2 (blue), ORFX initiation codon (red). **f** Relative replicon luciferase activities of deletion mutants (see Supplementary Fig. 13c for Huh7.5.1 cells). **g** Schematic design and results of experiment to quantify virus titer, RNA, and protein levels in released virions and infected cells. BSR cells were electroporated with pAVIC1-wt, -AUGm or -PTC1 T7 RNAs and incubated for 48 h. Clarified supernatants were titrated (first graph) or treated with RNase I and used for viral RNA isolation and subsequent quantification by qRT-PCR (second graph). Cells were collected, freeze-thawed three times and titrated on Caco2 cells (third graph). Total RNA from infected cells was isolated and virus gRNA quantified by qRT-PCR (fourth graph). Aliquots of electroporated cells were seeded on a 96-well plate, incubated for 48 h, fixed, permeabilized, stained with anti-CP antibody, and imaged by LI-COR followed by quantification using LI-COR software (fifth graph). **h** Cells were collected at 24 hpe, freeze-thawed three times and titrated on Caco2 cells. **i** Aliquots corresponding to $10^5$ and $10^4$ electroporated cells were seeded on a 96-well plate, fixed at 24 hpe, permeabilized, stained with anti-CP antibody, and imaged by LI-COR followed by quantification using LI-COR software. **j** Huh7.5.1 cells were electroporated with pAVIC1-wt, -AUGm or -PTC1 T7 RNAs and incubated for 24 h. Representative confocal images of fixed and permeabilized cells visualized for CP (green) and stained for nuclei (Hoechst, blue). Images represent z-stack projections. Scale bars are 10 μm. Graphs show means ± s.d. from n = 3 biologically independent experiments (**c, d, f, g, h**). Source data are provided as a Source Data file.

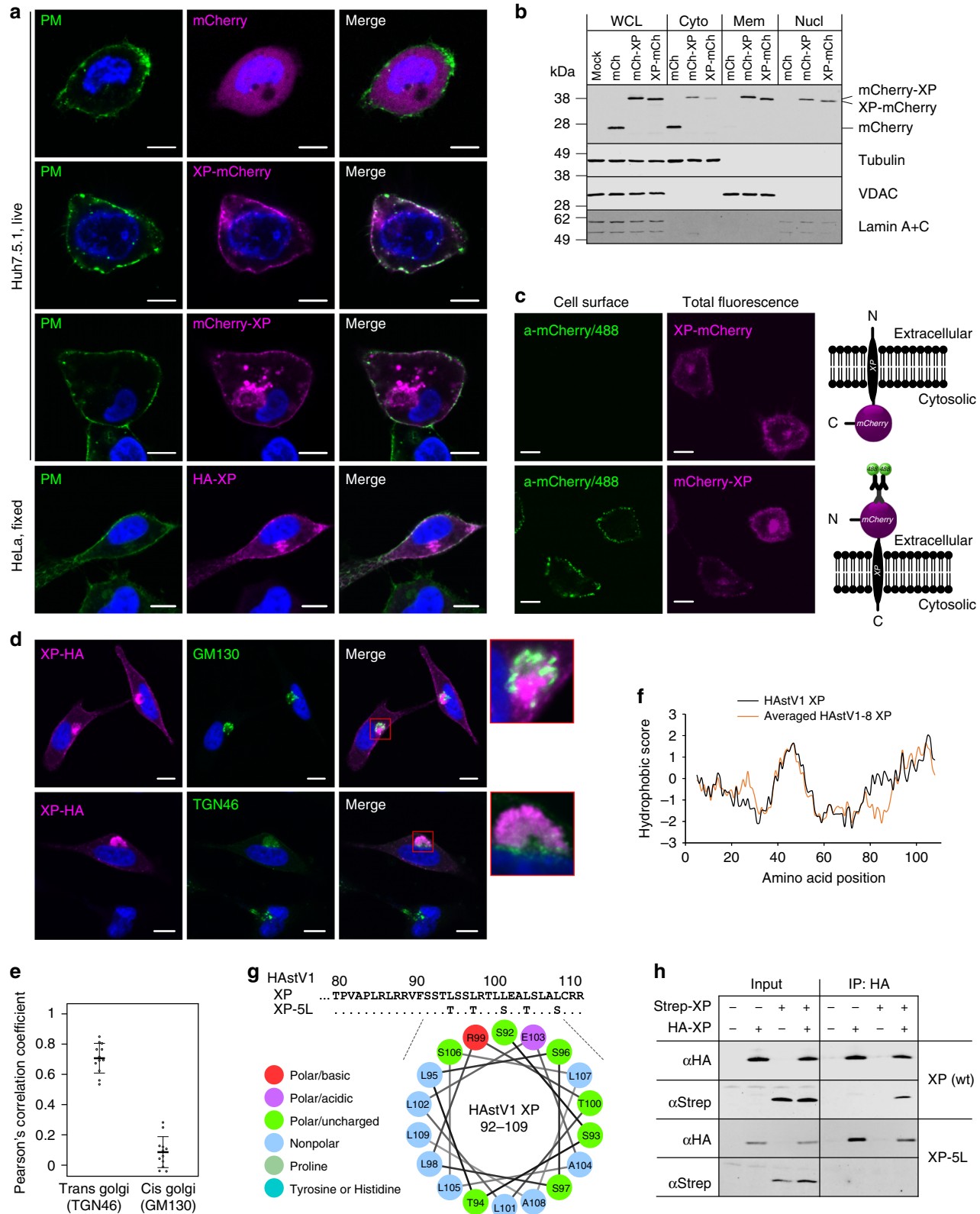

presence of a membrane-interacting domain in XP, which was not predicted by TM domain prediction software (Supplementary Fig. 8; see Methods). Since ORFX is completely embedded within ORF2, it may have greatly decreased evolutionary flexibility, perhaps resulting in the evolution of a non-canonical TM region. Some of the putative XP proteins encoded by other astroviruses do in fact have predicted TM regions (Supplementary Fig. 9). We also confirmed membrane and nuclear association of the XP-fused mCherry proteins by subcellular fractionation of transfected cells and subsequent analysis of the fractions (Fig. 5b and Supplementary Fig. 14). To investigate the potential topology of XP within the plasma membrane, we probed live electroporated HeLa

**Fig. 5 Cellular localization, membrane topology, and multimerization of XP. a** Huh7.5.1 cells were electroporated with pCAG-mCherry, pCAG-mCherry-XP, or pCAG-XP-mCherry. Representative confocal images of live cells stained for plasma membrane (WGA, green) and nuclei (Hoechst, blue); mCherry fluorescence is shown in red. HeLa cells were electroporated with pCAG-HA-XP, stained for plasma membrane (WGA, green), fixed, permeabilized and stained for XP with anti-HA antibody (red) and nuclei (Hoechst, blue). Images are averaged single plane scans. **b** HeLa cell lysates were fractionated and whole cell lysate (WCL), cytoplasmic (Cyto), membrane (Mem), and soluble nuclear (Nucl) fractions were analyzed by immunoblotting with antibodies to mCherry, tubulin, VDAC or Lamin A + C as indicated. See Supplementary Fig. 14 for complete images. **c** HeLa cells were electroporated with pCAG-XP-mCherry or pCAG-mCherry-XP. Cell surface mCherry on live cells was detected by incubation with anti-mCherry antibody, followed by staining with Alexa 488-labeled anti-rabbit IgG antibody and confocal microscopy. The images are averaged single plane scans. See Supplementary Fig. 15 for plasma membrane-permeabilized controls. A schematic representation of the observed membrane topologies is shown at right. **d** HeLa cells were electroporated with pCAG-XP-HA, fixed, permeabilized, and stained for XP (anti-HA, red), nuclei (Hoechst, blue), cis Golgi (anti-GM130, green), and trans Golgi (anti-TGN46, green). The images are averaged single plane scans. All scale bars are 10 μm (**a**,**c**,**d**). **e** Quantification of co-localization of XP-HA with TGN46 and GM130. The Pearson correlation coefficient was calculated for 12 images in each experiment. **f** Kyte-Doolittle hydropathy plots for HAstV XPs (see Supplementary Fig. 16 for individual plots). **g** C-terminal XP sequences for wt and 5L mutant HAstV1, and helical wheel representation of wt amino acids 92–109. **h** Huh7.5.1 cells were electroporated with wt or 5L mutant pCAG-Strep-XP or pCAG-HA-XP. At 16 hpe cells were lysed and subjected to immunoprecipitation using anti-HA magnetic beads. Presence of tagged proteins (5% of input and IP) was determined by western blotting using the indicated antibodies. See Supplementary Fig. 14 for complete images. Source data are provided as a Source Data file.

cells with anti-mCherry antibody, fixed the cells and analyzed them by confocal microscopy. We observed punctate staining across the plasma membrane when mCherry was fused to the N-terminus but not when it was fused to the C-terminus, suggesting an extracellular N-terminal topology (Fig. 5c). In contrast, for fixed and plasma membrane-permeabilized cells, anti-mCherry detected both N- and C-terminally tagged XP, confirming that the mCherry epitope was accessible on both XP fusions (Supplementary Fig. 15). The perinuclear localization suggested a link with Golgi stacks. Quantification of co-localization with cis and trans Golgi marker proteins demonstrated strong association of XP with the trans Golgi network (TGN) but not with the cis Golgi network (Fig. 5d–e).

Notwithstanding the TM predictions for HAstV XPs being below threshold (Supplementary Fig. 8), we hypothesized that the C-terminal region might still harbor a non-canonical TM domain. Additional support came from Kyte-Doolittle hydrophobicity profiles (Fig. 5f and Supplementary Fig. 16) and a C-terminal α-helical structure prediction for HAstV1–8 XPs (Supplementary Fig. 17). A helical wheel representation revealed an amphipathic region, including a penta-leucine hydrophobic stretch and a polar face (Fig. 5g). Given the above results, we hypothesized a possible viroporin function for XP. Viroporins are small hydrophobic virus-encoded ion channel proteins that modify cellular membranes to facilitate virus release from infected cells, and in some cases also play additional roles. We hypothesized that the C-terminal amphipathic α-helix might facilitate oligomerization to form TM pores.

To confirm multimerization, Huh7.5.1 cells expressing HA- and/or Strep-tagged XPs were lysed and subjected to immunoprecipitation with anti-HA-tag magnetic beads. Co-immunoprecipitation of HA- and Strep-tagged XPs was observed for lysates from cells expressing wt XP but not for cells expressing a mutant, XP-5L, where the penta-leucine stretch was mutated to serines and threonines (Fig. 5g–h). Taken together, these results recapitulate the situation for some well-studied viroporins, including the TGN-specific subcellular localization of HIV-1 Vpu[21], and the endoplasmic reticulum to TGN to plasma membrane trafficking of IAV M2[22].

**XPs from various astroviruses have a viroporin-like activity.** Pursuing the hypothesis of XP as a possible viroporin, we tested its ability to disturb membrane integrity leading to permeabilization when overexpressed in mammalian and bacterial cells. First, we utilized a previously described Sindbis virus replicon (SINV repC; Fig. 6a)[23] to overexpress XP or control proteins in

mammalian cells. This replicon has been demonstrated to be an extremely useful tool for investigating viroporins from different RNA viruses via induced permeabilization of the plasma membrane in BHK cells to the translation inhibitor hygromycin B (HB). BSR cells (a clone of BHK cells) were electroporated with in vitro transcribed SINV repC RNAs, and new protein synthesis was labeled with L-azidohomoalanine (AHA) in the presence or absence of HB, followed by lysis and click chemistry-based on-gel detection[24]. Expression of XP or Strep-tagged enterovirus 2B (a well-characterized viroporin[25]) led to almost total inhibition of protein synthesis (Fig. 6b–c) indicating that both proteins induced pronounced cell permeabilization to HB.

In addition to an amphipathic α-helix that facilitates oligomerization to form the TM pore, typical viroporins also possess adjacent positively charged residues that anchor the viroporin in the membrane[26,27]. To test these features in XP, two conserved RR motifs present in all eight HAstV serotypes (Fig. 1e) were mutated to alanines. Both anchoring RR (Fig. 6d) and amphipathic 5L (Fig. 5g) mutations resulted in loss of XP activity in this system (Fig. 6c). Thus, the C-terminal domain of XP is involved in membrane permeabilization and harbors key features of a viroporin.

To investigate conservation of XP activity across different astrovirus species, we tested several other astrovirus XPs using the mammalian permeabilization assay. We used XPs from closely related HAstV4, feline (FAstV; genotype Ia), canine (CAstV; genotype Ib), and more distantly related porcine (PAstV4; genotype IIIc) astroviruses in this experiment (Fig. 6e). Similar to HAstV1, all these XPs contain one or two predicted amphipathic α-helices (Supplementary Fig. 18). All tested XPs resulted in increased cell permeabilization to HB (Fig. 6c) confirming conservation of XP activity when tested with this assay.

Another widely used assay to assess the ability of proteins to permeabilize cellular membranes is based on impaired growth of *Escherichia coli* upon induced overexpression of a membrane-permeabilizing protein[26,28]. Consistent with the results observed in the mammalian system, induced expression of each of the four XPs or the enteroviral viroporin 2B in *E. coli* resulted in cytotoxicity and impaired growth, whereas the 5L and RR2 (but not RR1) XP mutations abrogated this effect (Supplementary Fig. 19).

To test the importance of the viroporin-like features of XP in the context of astrovirus infection, a pAVIC1-based 5L mutant virus was generated, using mutations that did not alter the CP amino acid sequence (5L to Q/S/Q/H/Q; Fig. 6f). Similar to the PTC1 mutant, the 5L mutant virus had a strong reduction in

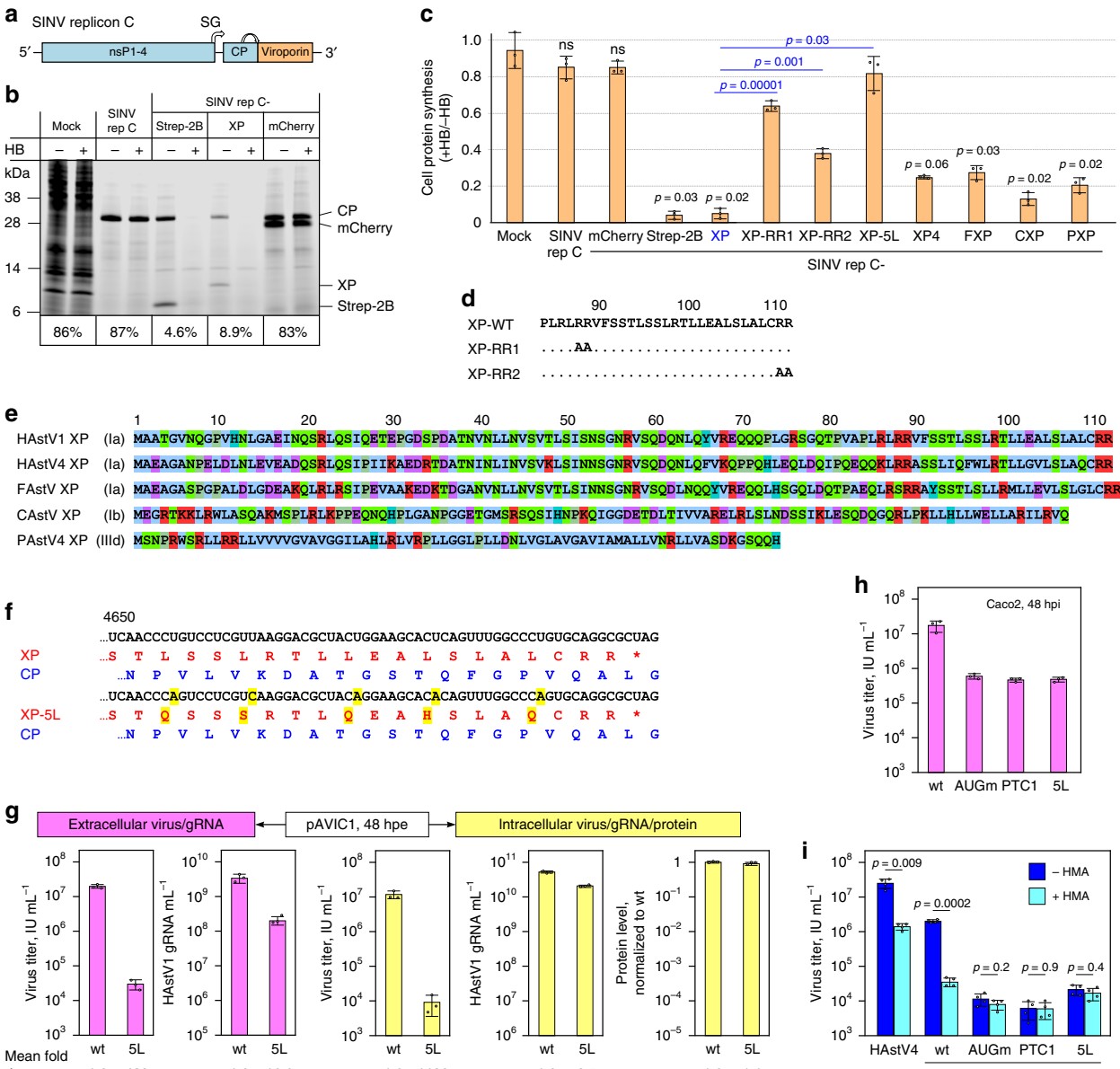

**Fig. 6 XPs from different astroviruses have viroporin-like activity. a** Schematic representation of SINV replicon C used to evaluate cell-permeabilizing activity of expressed proteins. SINV elements: nsP1-4 – non-structural polyprotein; CP – capsid protein; SG – subgenomic promoter. **b** Membrane permeabilization in BSR cells at 8 h post RNA electroporation with Sindbis virus replicons (SINV repC) expressing HAstV1 XP, enterovirus Strep-2B (positive control), or mCherry (negative control). Ongoing protein synthesis was labeled with 1 mM AHA in the presence or absence of 1 mM HB as a translation inhibitor. Cells were lysed and AHA-bearing proteins were ligated to the fluorescent reporter IRDye800CW Alkyne by click chemistry, separated by SDS-PAGE, and visualized by in-gel fluorescence. Numbers below each pair of samples indicate protein synthesis quantified for HB-treated cells relative to values obtained for untreated cells. **c** Statistical analysis of membrane permeabilization caused by different XPs and mutants and the indicated control proteins in BSR cells. Bars indicate the amount of protein synthesis in HB-treated cells relative to untreated cells. *P* values are from comparisons with mock (BSR; black) or with pSINV-repC-XP (blue). **d** Sequences of wt, RR1, and RR2 HAstV1 mutants. **e** Sequences of HAstV1, HAstV4, and feline (FAstV), canine (CAstV) and porcine (PAstV) astrovirus XPs. **f** Alignment of pAVIC1-wt and -5L showing translation of XP (red) and the overlapping CP (blue). Introduced nucleotide and amino acid changes are highlighted in yellow. **g** Quantification of virus titer, RNA, and protein levels in released virions and infected cells as described in Fig. 4g. **h** Caco2 cells were infected with the indicated viruses at MOI 0.2 and incubated for 48 h. Released virus in clarified supernatants was titrated. **i** Caco2 cells were infected with the indicated viruses at MOI 0.2 in the presence (dark blue) or absence (light blue) of 5 μM hexamethylene amiloride (HMA). Intracellular (HMA-free) virus was titrated. See Supplementary Fig. 20 for cell toxicity data. *P* values come from two-tailed *t*-tests without adjustment for multiple comparisons. Error bars indicate mean ± s.d.; *n* = 3 (**c**, **g**, **h**) or 4 (**i**) biologically independent experiments. Source data are provided as a Source Data file.

released virus (439-fold), but not in intracellular levels of viral RNA and protein (Fig. 6g). Similar to the PTC1 mutant, the 5L mutant was also impaired in virus release in infected Caco2 cells (Fig. 6h), consistent with the viroporin-like activity of XP being important in virion formation and/or release.

Finally, we tested several viroporin inhibiting antiviral drugs for ability to inhibit astrovirus infection. Hexamethylene amiloride (HMA)— demonstrated to inhibit HCV p7, HIV-1 Vpu, and coronavirus E viroporin channel activity[27,28]—was found to specifically inhibit wt HAstV1 and HAstV4 infection,

but not HAstV1-AUGm, -PTC1, or -5L mutant virus infection when used at noncytotoxic concentrations in Caco2 cells (5 μM; Fig. 6i and Supplementary Fig. 20).

Taken together, these findings indicate that the astrovirus XP protein has a viroporin-like function, which is directly involved in the virus replication cycle.

## Discussion

The data presented here demonstrate the existence of an additional protein, XP, encoded within the human astrovirus genome. XP is important for virus growth, localizes to the TGN and plasma membranes, and plays a role in virion formation and/or release. Of note, previous structural studies on the HAstV CP[29] and virion[30] have not indicated the presence of any additional proteins, such as XP, within the virion itself.

Viroporins have been reported for many enveloped and non-enveloped viruses and—although they can play roles in virus entry and modulation of cellular pathways—most often they facilitate virus assembly and release[28]. However, no viroporin candidate had been previously predicted for astroviruses. Identifying a viroporin is challenging due to the lack of homology among viroporins from different viruses. Viroporins are typically small hydrophobic integral membrane proteins of around 100 aa in size, with two motifs—an amphipathic TM α-helix and an adjacent cluster of positively charged residues; mutation of these residues generally abolishes viroporin activity[26]. Using different assays and computational approaches, we show that XP fulfils these criteria. XP is capable of permeabilizing cellular membranes and has a distinct N-terminal extracellular topology with one TM domain. Future work will be needed to confirm potential ion channel activity, and characterize ion specificity, oligomeric state and structural organization, and any additional processes affected by XP expression in the context of viral infection. The localization of XP not only at the plasma membrane but also in the perinuclear TGN membranes raises the possibility that XP may also have additional functions. Whereas the C-terminal α-helix appears to be associated with the cell-permeabilizing activity, additional functions (if any) of the extended N-terminal domain of XP remain to be studied. Notably, other viroporins have additional functions ranging from membrane remodeling to generate viral replication sites, subversion of autophagy, induction of endoplasmic reticulum stress, disruption of calcium homeostasis, and regulation of innate immune and pro-inflammatory responses[27,28,31]. In astroviruses, pathology is often associated with increased gut epithelial permeability, disruption of the tight-junction protein occludin, and other cell perturbations in the absence of cell death;[32,33] however the involvement of CP, XP and other proteins in these processes requires further investigation.

Comparative genomic analysis suggests that the presence of a protein-coding ORF overlapping ORF2 is widespread in mammalian astroviruses. It appears to be almost ubiquitous in genogroups I, III, and IV and is probably present in the single genogroup V sequence. It also appears to be present in various unassigned sequences and clades. ORFX frequently has a hydrophobic stretch that is often predicted to be a TM domain and in other cases may represent a non-canonical TM domain refractory to detection with standard TM-predicting software (as appears to be the case for HAstV1). While normally absent from genogroup II astroviruses, an ORFX appears to be present in one clade (herein referred to as IId), whereas in the clade referred to herein as IIc we predicted an ORFY in the −1 instead of +1 frame, and accessed via ribosomal frameshifting instead of leaky scanning. Given the sporadic appearance of an overlapping ORF across the genogroup II phylogeny, it seems likely that the

genogroup II ORFX and ORFY evolved independently from ORFX in genogroups I, III, and IV, and may also therefore have different functions. In contrast, these other ORFXs may or may not have a common ancestor and/or common function (it is unclear to what extent genogroups I, III, and IV form a monophyletic group; Supplementary Figs 1 and 2). The N-terminal ~70 aa of CP are dispensable for particle assembly and likely structurally disordered[29] and may therefore be evolutionarily fairly flexible. Thus, this region of ORF2 may be unusually tolerant to the coding constraints imposed by overlapping genes. Together with the high translation level of sgRNAs at later timepoints (Fig. 2a), and the ease with which 5′ proximal ORFs can be expressed (requiring only leaky scanning rather than more complex expression mechanisms such as internal ribosome entry or ribosomal frameshifting[10]), the 5′ region of the sgRNA may be particularly well-suited to the evolution of an overlapping gene, consistent with multiple independent origins of ORFXs. While XP appears to be a very important protein for HAstV1, the different genogroups of astroviruses are highly divergent from each other and it would not be surprising if astroviruses without XP have evolved different routes to substitute or complement its function.

During this study we also developed and characterized an astrovirus replicon system. This will be of broad utility to the astrovirus research community. Astroviruses are one of the major causes of infant gastroenteritis; they are widespread among mammals; and non-classical human-infecting astroviruses (such as the MLB and VA/HMO clades) have recently been recognized. Nonetheless, despite their ubiquity and importance, astroviruses represent some of the least well-studied human viruses, partly because it has been difficult to establish efficient lab systems to study them. The replicon system developed herein will now permit detailed characterization of astrovirus replication and gene expression, and facilitate research into antiviral drugs.

In summary, using comparative genomic analyses we predicted an additional gene in the *Mamastrovirus* genus; using ribosome profiling we demonstrated XP expression in HAstV1-infected cells; and using various techniques we demonstrated the crucial role of XP in virus growth by promoting virus assembly and/or release, which is likely associated with its viroporin-like activity. XP exhibits parallels with other well-studied viroporins such as HIV-1 Vpu, coronavirus E, and IAV M2 in localization, cell permeabilization activity, and drug inhibition. These findings add an additional dimension to astrovirus molecular biology, with potential impacts for new therapeutics (e.g., compounds such as HMA that block XP activity) and vaccine development (e.g., by inhibiting XP expression).

## Methods

**Comparative genomic analysis**. Mammalian astrovirus nucleotide sequences were downloaded from NCBI on 26 July 2018. Patent sequence records and sequences with ≥20 ambiguous nucleotide codes (e.g., N) were removed. For the full-genome analyses, only sequences covering all or nearly all of ORF1a, ORF1b, and ORF2 were retained, giving 221 sequences (listed in Supplementary Fig. 1). For the ORF2 analyses, only sequences covering all or nearly all of ORF2 were retained, giving 415 sequences (listed in Supplementary Table 2). To identify the correct 5′ end of ORF1b, we identified the AAAAAAC frameshift site. To identify the correct initiation site of ORF2, we identified the highly conserved sgRNA promoter nucleotides[34] and selected the next ORF2-frame AUG codon as the ORF2 start site in representative reference sequences; for the other sequences, the ORF2 start site was identified by amino acid alignment to one of the reference sequences. ORF1b and ORF2 sequences were extracted, translated to amino acid sequences, aligned with MUSCLE version 3.8.31[35], and maximum likelihood phylogenetic trees were estimated using the Bayesian Markov chain Monte Carlo method implemented in MrBayes version 3.2.3[36] sampling across the default set of fixed amino acid rate matrices, with 1,000,000 (Supplementary Figs. 1 and 2) or 5,000,000 (Supplementary Fig. 4) generations, discarding the first 25% as burn-in (other parameters were left at defaults). Trees were visualized with FigTree version 1.4.2 (http://tree.bio.ed.ac.uk/software/figtree/). Since not all sequences have formal species names,

we named virus sequences in trees according to the ORGANISM field of the respective GenBank record.

The 221-sequence ORF1b tree was used to manually select clades (Fig. 1a) for full-genome SYNPLOT2 analyses (Supplementary Fig. 3). For the ORF2-only SYNPLOT2 analyses (Supplementary Fig. 5), we used a more objective method to select clades. Through an iterative procedure of clustering the 415 ORF2 sequences based on amino acid identity to a set of reference sequences, and selecting sequences distal from all reference sequences as new reference sequences, we arrived at a set of 42 ORF2 reference sequences: KT946734, Z25771, JN420356, KP404149, JX556691, AY179509, Y15937, FJ973620, JF713713, JX544743, JF713710, LC047798, FJ222451, JX556693, KT946736, KP663426, KY855439, JF729316, KY855437, KY024237, MG693176, JF755422, EU847155, KT946731, LC047794, KF787112, JN420352, JN420359, LC047787, KT963069, HQ916316, KX645667, KT963070, FJ890355, HQ668129, HQ668143, FJ571066, EU847145, EU847144, FJ571067, GQ415660, FJ571068. The highest reference to reference pairwise CP amino acid identity is 0.5173; no non-reference has <50% identity to a reference, and only 20 have <55% identity; 82 and 4 sequences have >50% and >55% identity, respectively, to >1 references. Each non-reference sequence was then clustered with the reference sequence to which it has highest CP amino acid identity. This resulted in 16 singleton clusters and 26 multi-sequence clusters.

Synonymous site conservation was analyzed with SYNPLOT2 version 1[9]. For the full-genome analyses we generated codon-respecting alignments using the same procedure as in previous work[9]. In brief, each individual genome sequence was aligned to a reference sequence using code2aln version 1.2[37]. Genomes were then mapped to reference sequence coordinates by removing alignment positions that contained a gap character in the reference sequence, and these pairwise alignments were combined to give the multiple sequence alignment. To assess conservation at synonymous sites, the ORF1a, ORF1b, and ORF2 coding regions were extracted from the alignment (with codons selected from the longer ORF in each overlap region), concatenated in-frame, and the alignment analyzed with SYNPLOT2 using a 25-codon sliding window. Conservation statistics were then mapped back to reference genome coordinates for plotting. For the ORF2-only SYNPLOT2 analyses, any duplicate sequences were removed and the remaining ORF2 sequences in each clade were translated, aligned using MUSCLE as amino acid sequences, back-translated to codon-respecting nucleotide alignments, and the alignment analyzed with SYNPLOT2 as above. In contrast to the full-genome alignments, all alignment gaps were retained instead of mapping to a specific reference sequence coordinate system.

Calculation of the pI and molecular mass of XP peptides and other sequence processing were performed with pepstats and other programs from EMBOSS version 6.6.0.0[33]. TM domains were predicted with Phobius (http://phobius.sbc.su.se; Apr 2019)[38]. The XP proteins of HAstVs 1–8 were additionally queried with TMHMM (http://www.cbs.dtu.dk/services/TMHMM/; Feb 2020; weak TM prediction for HAstV3, no TM predicted for other HAstVs) and SOSUI (http://harrier.nagahama-i-bio.ac.jp/sosui/sosui_submit.html; Feb 2020; no TMs predicted). XP secondary structures were predicted with RaptorX (http://raptorx.uchicago.edu/; May 2019)[39]. Kyte-Doolittle hydropathy plots were calculated with protscale (https://web.expasy.org/protscale; May 2019). Helical wheel representations were created using Heliquest (http://heliquest.ipmc.cnrs.fr; May 2019). To search for potential homologues of XP, XPs from Supplementary Figs. 6d, 7 and 9 were queried with HHpred (https://toolkit.tuebingen.mpg.de/tools/hhpred; Jun 2019)[40]. Only one of the 33 queries retrieved a match with an expectation value < 1—namely murine astrovirus (genogroup IIIa), which obtained an expectation value 0.38 hit to a fragment of Protein Data Bank accession 5NIK [https://www.rcsb.org/structure/5NIK], an E. coli outer membrane ion channel protein (most likely a manifestation of convergent evolution).

**Cells and viruses.** BSR (single cell clone of BHK-21 cells) and HeLa cells (ATCC, CCL-2) were maintained at 37 °C in DMEM supplemented with 10% fetal bovine serum (FBS), 1 mM L-glutamine, and antibiotics. Caco2 and Huh7.5.1 cells[41] (Apath, Brooklyn, NY) were maintained in the same media supplemented with non-essential amino acids. All cells were mycoplasma tested (MycoAlert™ PLUS Assay, Lonza); BSR, Huh7.5.1 and Caco2 cells were also tested by deep sequencing.

The infectious clone of HAstV1 (pAVIC1, GenBank accession number L23513.1 [https://www.ncbi.nlm.nih.gov/nuccore/L23513.1]) was developed previously[17]. The reverse genetics procedure was compiled from several previously published approaches[17,18]. Initial virus was recovered from T7 transcribed RNA using reverse transfection of Huh7.5.1 cells by Lipofectamine® 2000 (Invitrogen) (for virus rescue; see Fig. 3) or electroporation of BSR cells in PBS at 800 V and 25 µF using a Bio-Rad Gene Pulser Xcell™ electroporation system (see Fig. 4 for an analysis of virus RNA in cells versus released particles). For virus passaging, the collected supernatant was treated with 10 µg mL⁻¹ trypsin (Type IX, Sigma, #T0303) for 30 min at 37 °C, diluted five times with serum-free media, and used for infection of Caco2 cells. After 3 h of incubation, the virus containing media was replaced with serum-free media containing 0.6 µg mL⁻¹ trypsin, and cells were incubated for 48–72 h until appearance of CPE. After three freeze-thaw cycles, viral stocks were aliquoted, frozen, and stored at −70 °C. Viral stocks were titrated by immunofluorescence-based detection with 8E7 astrovirus antibody (1:750 dilution)[42], but using infrared detection readout, combined with automated LI-COR software-based quantification.

**Virus evolution.** Virus evolution was performed in Caco2 cells in triplicate by passaging mutant viruses at different dilutions (corresponding to MOIs ranging from 0.001 to 0.1). Since HAstVs do not form plaques in Caco2 cells, an end-point dilution approach was used instead of plaque purification. Infections that resulted in CPE by 5 dpi were used for subsequent passage at a similar dilution and incubation time. Following an increase of at least tenfold over P1 titer (Fig. 3c), passages in individual flasks with infected Caco2 cells were used for RNA isolation by Direct-zol RNA MicroPrep (Zymo research), RT-PCR, and Sanger sequencing of the fragment containing the mutated region of the virus genome.

**Plasmids.** For mammalian expression of XP, the coding sequence of mCherry alone or HAstV1 XP fused to sequence encoding mCherry, HA or Strep tag at either the N- or C-terminus was inserted into vector pCAG-PM[43] using AflII and PacI restriction sites. The resulting constructs—designated pCAG-mCherry, pCAG-XP-mCherry, pCAG-mCherry-XP, pCAG-XP-HA, pCAG-HA-XP, and pCAG-Strep-XP—were confirmed by sequencing.

All virus genome mutations (Supplementary Fig. 11 and Fig. 6f) were introduced using site-directed mutagenesis of pAVIC1 and confirmed by sequencing. The resulting plasmids were linearized with XhoI prior to T7 RNA transcription. To create the HAstV1 replicon system, the pAVIC1 infectious clone was left intact up to the end of ORFX, then followed by a foot-and-mouth disease virus 2A sequence and a Renilla luciferase (RLuc) sequence fused in either the ORF2 (pO2RL) or ORFX (pXRL) reading frame, followed by the last 624 nt of the virus genome and a poly-A tail (Fig. 4a; GenBank accession number MN030571 [https://www.ncbi.nlm.nih.gov/nuccore/MN030571.1/]). All mutations were introduced from the corresponding pAVIC1 mutants using available restriction sites and all constructs were confirmed by sequencing. The resulting plasmids were linearized with XhoI prior to T7 RNA transcription.

For bacterial expression of XP and related control proteins (mCherry and N-terminally Strep-tagged enterovirus 2B from echovirus 7), the relevant coding sequences were inserted into the pOPT expression plasmid[44] between NdeI and BamHI restriction sites. Feline (KF374704.1), porcine (LC201600.1), and canine (FM213332.1) XP coding sequences were commercially synthesized (Integrated DNA Technologies). Sequence encoding HAstV4 XP was RT-PCR amplified from a clinical HAstV4 isolate kindly provided by Susana Guix (University of Barcelona, Spain). Each XP coding sequence was inserted into the pOPT plasmid with a C- or N-terminal Strep-tag as indicated (Supplementary Fig. 18). For the permeabilization assay, Sindbis virus derived replicons (SINV repC) expressing one of the XPs, mCherry or Strep-tagged enterovirus 2B from echovirus 7 were created by cloning of the potential viroporin-encoding sequence downstream of the alphavirus CP gene[23].

**Ribosome profiling.** Caco2 cells were grown on 150 mm dishes to reach 80–90% confluency. The cells were infected at MOI 5 with HAstV1 virus stock (passage 2, derived from pAVIC1 T7 RNA). At 12 hpi, cells were either not treated (NT) or treated with 50 µM LTM for 30 min, flash frozen in a dry ice/ethanol bath, and lysed in the presence of 0.36 mM cycloheximide. Cell lysates were subjected to Ribo-Seq based on the previously described protocols[14,45,46], except the Ribo-Zero Gold rRNA removal kit (Illumina), not DSN, was used to deplete ribosomal RNA. Amplicon libraries were deep sequenced using an Illumina NextSeq platform.

**Computational analysis of Ribo-Seq data.** Similar to previous work[14], the Ribo-Seq analysis was performed as follows. Adapter sequences were trimmed using the FASTX-Toolkit version 0.0.13 (http://hannonlab.cshl.edu/fastx_toolkit) and trimmed reads shorter than 25 nt were discarded. Reads were mapped to host (Homo sapiens) and virus RNA using bowtie1 version 0.12.9[47], with parameters -v 2 --best (i.e., maximum two mismatches, report best match). Mapping was performed in the following order: host rRNA, virus RNA, host RefSeq mRNA, host non-coding RNA, host genome.

To normalize for library size, reads per million mapped reads (RPM) values were calculated using the sum of positive-sense virus and host RefSeq mRNA reads as the denominator. A +12 nt offset was applied to the RPF 5′ end positions to give the approximate ribosomal P-site positions. To calculate the phasing and length distributions of host and virus RPFs, only RPFs whose 5′ end (+12 nt offset) mapped between the 13th nucleotide from the beginning and the 18th nucleotide from the end of coding sequences (ORF1a, ORF1b and ORF2 for HAstV; RefSeq mRNAs for host) were counted, thus avoiding RPFs near initiation and termination sites. For Supplementary Fig. 10, the dual-coding region where ORFX overlaps ORF2 was also excluded. Histograms of host RPF positions (5′ end +12 nt offset) relative to initiation and termination sites were derived from RPFs mapping to RefSeq mRNAs with annotated coding regions ≥450 nt in length and with annotated 5′ and 3′ UTRs ≥60 nt in length. Virus ORF1a, ORF1b, and ORF2 ribosome densities (used herein as a proxy for translation levels) were calculated by counting RPFs whose 5′ end (+12 nt offset) mapped within the regions 101–2782, 2866–4311 or 4730–6676, respectively (i.e., excluding the dual-coding regions and excluding reads with P-sites mapping within 15 nt of initiation, termination or ribosomal frameshifting sites).

To compare phasing in the region of ORF2 overlapped by ORFX with the region of ORF2 not overlapped by ORFX (Fig. 2c) we counted RPFs whose 5′ end

(+12 nt offset) mapped within the regions 4388–4693 or 4727–6673, respectively. We then compared the fraction of RPFs in phases 1, 2, and 3 in the ORF2/ORFX overlap region $[v_1, v_2, v_3]$ with the corresponding fractions for host mRNAs $[m_1, m_2, m_3]$, leading to the three equations $[v_1, v_2, v_3] \approx c \times [m_1, m_2, m_3] + x \times [m_3, m_1, m_2]$, where $c$ and $x$ are the relative expression levels of ORF2 and ORFX, respectively, and $c + x = 1$. Writing $[q_1, q_2, q_3] = c \times [m_1, m_2, m_3] + (1 - c) \times [m_3, m_1, m_2] - [v_1, v_2, v_3]$, we found $c$ to minimize $(q_1^2 + q_2^2 + q_3^2)^{1/2}$, giving $c = 0.79$ and 0.78 for repeats 1 and 2 respectively. Thus the estimated ORF2:ORFX expression ratios are 0.79:0.21 and 0.78:0.22, i.e., ORFX is expressed at ~27% of the level of ORF2. For bootstrap resampling of the ORF2/ORFX overlap region and the nonoverlapping region of ORF2, each resampling was of 102 codons chosen at random (with replacement) from the relevent region, 4388–4693 or 4727–6673 respectively (ORFX has 112 codons but—see above—we exclude RPFs with P-sites mapping within 15 nt of initiation or termination sites). For each resampling, we found the estimated XP-frame:CP-frame expression ratio as above.

**Permeabilization assay**. BSR cells electroporated with RNA synthesized in vitro from the different linearized replicon constructs or with transcription buffer alone were seeded in 6-well plates. At 8 hpe, cells were pretreated with 1 mM HB (Invitrogen) in methionine-free media (Gibco, Life Technologies) for 20 min or left untreated. Next, ongoing protein synthesis was labeled with 1 mM AHA (Invitrogen) in methionine-free media in the presence or absence of 1 mM HB for 40 min. Finally, cells were lysed in 1% SDS containing PBS supplemented with protein inhibitor cocktail (Roche) and 250 U mL$^{-1}$ bensonase (Sigma). Labeled proteins were ligated with 25 µM IRDye800CW Alkyne (LI-COR) in the presence of CuSO$_4$ (CCT, 100 µM), tris-hydroxypropyltriazolylmethylamine (THPTA, CCT, 500 µM), aminoguanidine (Cambridge Bioscience, 5 mM), and sodium L-ascorbate (Sigma, 2.5 mM) for 2 h at room temperature. The final products were lysed, resolved in Novex 10–20% tricine protein gels (Invitrogen), fixed in 50% ethanol and 10% acetic acid for 10 min, scanned on LI-COR and analyzed by densitometry using LI-COR software. Experiments were repeated three times.

***E. coli* lysis assay**. For the assessment of protein viroporin-like activity in *E. coli*, BL21(DE3)pLysS cells (Promega) were transformed with pOPT constructs, grown in the presence of ampicillin (100 µg mL$^{-1}$) at 37 °C to an optical density at 600 nm (OD$_{600}$) of 0.4–0.6, and then 1 mM isopropyl-β-D thiogalactopyranoside (IPTG) was added to induce protein expression. Subsequently, optical densities were measured for induced and non-induced samples in triplicate over a time course of 180 min post induction using a Spectra Max i3x (Molecular Devices) microplate reader. Non-induced, and 60 and 120 min post induction samples were also collected for protein detection by western blot.

**Fractionation analysis**. To analyze the cellular distribution of overexpressed XP fusions, electroporation of HeLa cells was performed in full media at 240 V and 975 µF using a Bio-Rad Gene Pulser. At 20 hpe, cells were washed with PBS and fractionated using a subcellular protein fractionation kit for cultured cells (Thermo Scientific) according to the manufacturer's instructions. Equal aliquots of whole cell lysate, cytoplasmic, membrane and soluble nuclear fractions were analyzed by western blot using the indicated target-specific antibodies: anti-mCherry (Abcam, ab167453, 1:1000), anti-tubulin (Abcam, ab15568, 1:500), anti-VDAC1 (Abcam, ab14734, 1:1000), and anti-LAMIN A + C (Abcam, ab133256, 1:3000).

**Co-immunoprecipitation**. To analyze multimerization of overexpressed XP, electroporation of Huh7.5.1 cells was performed in full media at 220 V and 975 µF using a Bio-Rad Gene Pulser. At 20 hpe, cells were washed with PBS and co-immunoprecipitation was performed using the Pierce™ Magnetic HA-Tag IP/Co-IP Kit (Thermo Scientific) according to the manufacturer's instructions. Aliquots of input (5%) and immunoprecipitation fractions were analyzed by western blot using the indicated tag-specific antibodies: anti-HA (Abcam, ab20084, 1:1000) and anti-Strep (Abcam, ab184224, 1:1000).

**Immunofluorescence microscopy**. For the analysis of intracellular localization of overexpressed XP fusions, electroporation of Huh7.5.1 or HeLa cells was performed as described in the previous section. At 16 hpe, cells were fixed with 4% paraformaldehyde (PFA) for 20 min at room temperature, followed by permeabilization with PBS containing 0.1% Triton X-100 for 10 min. Nuclei were counter-stained with Hoechst (Thermo Scientific). For live cell imaging, plasma membranes were stained with Wheat Germ Agglutinin Alexa Fluor™ 488 Conjugate (WGA, Thermo Scientific, 1:250) for 20 min. For live cell probing, at 16 hpe cells were washed with cold PBS and incubated with anti-mCherry antibody (Abcam, ab167453, 1:500 dilution) on ice for 1 h, then washed with PBS, fixed with 4% PFA for 20 min at room temperature and incubated with secondary antibody (Alexa Fluor 488-conjugated goat anti-rabbit, Thermo Fisher, A21441). To confirm that anti-mCherry antibody can recognize both XP fusions, transfected cells were fixed with 4% PFA, washed with KHM buffer (110 mM potassium acetate, 3 mM MgCl$_2$, 20 mM Hepes pH 7.2), plasma membrane was selectively permeabilized with 30 µM digitonin in KHM buffer, and cells were stained with anti-mCherry antibody followed by detection using secondary antibody as above. For Golgi apparatus imaging, fixed and permeabilized cells were stained with TGN46 (AHP500G, BioRad,

1:200) and GM130 (610882, BD Biosciences, 1:200) antibodies followed by Alexa Fluor 488-conjugated secondary antibody staining (Thermo Fisher; A21441 and A11015, respectively, 1:2000). HA-tagged XP was imaged using anti-HA tag antibody (Abcam, ab20084, 1:1000) followed by secondary antibody (Alexa Fluor 568-conjugated donkey anti-rabbit, Thermo Fisher, A10042, 1:1000). The images are single-plane images taken with a Leica SP5 Confocal Microscope using a water-immersion 63× objective.

For the analysis of CP localization in Huh7.5.1 cells, cells electroporated with pAVIC1 or mutants were incubated for 24 h, fixed and permeabilized as above. CP was detected using Astrovirus 8E7 antibody (Santa Cruz Biotechnology, sc-53559, 1:750) followed by incubation with secondary 488-conjugated antibody. The images are z-stack projection taken with a Leica SP5 Confocal Microscope using a water-immersion 63× objective.

The extent of co-localization was quantified by calculating the Pearson correlation coefficient (PCC) using the JACoP plugin of ImageJ (https://imagej.nih.gov/ij/index.html). The PCC measures the pixel-by-pixel covariance of the signal levels of two images. A PCC of 1 indicates perfect co-localization, 0 no correlation, and −1 perfect anti-correlation. The PCC was calculated for 12 images in each experiment.

**Analysis of cellular and released virus samples**. BSR cells were electroporated with in vitro transcribed RNAs derived from wt or mutant pAVIC1 using a double-pulse protocol in a BioRad Gene Pulser at 800 V and 25 µF in PBS. After 4 hpe, cells were washed three times and supplemented with virus production serum-free media (VP-SFM, Gibco) containing 1 mM L-glutamine and 0.6 µg mL$^{-1}$ trypsin. At 48 hpe media samples were collected and centrifuged at 9600 g for 5 min. The released virus samples were titrated on Caco2 cells as described above. For RT-qPCR analysis, a 150 µl aliquot of each sample was mixed with $4 \times 10^6$ plaque forming units of purified Sindbis virus (SINV) stock, which was used for normalization and to control the quality of RNA isolation. The virus samples were treated with 300 units of RNase I (Ambion) for 30 min followed by SUPERaseI RNase Inhibitor (Thermo Fisher) to eliminate RNA contaminants outside of virus particles. RNA was then extracted using the Qiagen QIAamp viral RNA mini kit. Reverse transcription was performed using the QuantiTect reverse transcription kit (Qiagen) using virus-specific reverse primers for SINV (SINV_RT) and HAstV1 (HAstV1-RT) (a complete list of primers used in this study is given in Supplementary Table 3). HAstV1 (HAstV1-qF and HAstV1-qR) and SINV (SINV-qF and SINV-qR) specific primers were used to quantify corresponding virus RNAs; the primer efficiency were within 95–105%. Quantitative PCR was performed in triplicate using SsoFast EvaGreen Supermix (Bio-Rad) in a ViiA 7 Real-time PCR system (Applied Biosystems) for 40 cycles with two steps per cycle. Results were normalized to the amount of SINV RNA in the same sample. Fold differences in RNA concentration were calculated using the $2^{-\Delta\Delta CT}$ method.

For analysis of virus RNA present in infected cells, cells were pelleted at 9600 g for 5 min and total RNA was extracted using Direct-zol RNA MiniPrep Plus (Zymo Research) according to the manufacturer's protocol. Reverse transcription was performed using the QuantiTect reverse transcription kit (Qiagen) using random and HAstV1-specific reverse primers. For qPCR, hamster GAPDH-specific (hamGAPDH-qF and hamGAPDH-qR) and HAstV1-specific primers (above) were used. Results were normalized to the amount of GAPDH RNA in the same sample. Fold differences in RNA concentration were calculated using the $2^{-\Delta\Delta CT}$ method.

Intracellular virus was quantified at 24 and 48 hpe. Cells were pelleted, resuspended in serum-free media, freeze-thawed three times, and titrated on Caco2 cells as described above. For HAstV1 CP quantification, electroporated cells were seeded on 96-well plates at different dilutions, and at 24 hpe fixed with 4% PFA and permeabilized with 0.5% triton X-100 in PBS. CP was detected using Astrovirus 8E7 antibody followed by LI-COR visualization and quantification.

The analysis of cellular cytotoxicity was performed using the CyQUANT LDH cytotoxicity assay (Thermo Scientific). Leaked cytoplasmic enzyme LDH in cell culture supernatants was quantified after enzymatic conversion, and absorbance was measured at 490 nm in a Spectra Max i3x (Molecular Devices) microplate reader according to the manufacturer's instructions.

**SDS-PAGE and immunoblotting**. Lysates from the above mentioned assays were analyzed by SDS-PAGE, using standard 12% SDS-PAGE to resolve mCherry and its XP-fusion variants, and precast Novex™ 10–20% tricine protein gels (Thermo Fisher) to resolve XPs and enterovirus 2B. Proteins were then transferred to 0.2 µm nitrocellulose membranes and blocked with 4% Marvel milk powder in phosphate-buffered saline (PBS). Immunoblotting of mCherry was performed using anti-mCherry antibody (Abcam, ab167453, 1:3000). A custom rabbit polyclonal antibody raised against XP peptide SNSGNRVSQDQNLQ (GenScript; only able to detect strongly overexpressed XP, 1:250) and an anti-Strep mouse antibody (Abcam, ab184224, 1:1000) were used for detecting HAstV1 XP and Strep-tagged proteins, respectively. The following antibodies were used for cellular targets: anti-tubulin (Abcam, ab15568, 1:500), anti-VDAC1 (Abcam, ab14734, 1:1000), and anti-LAMIN A + C (Abcam, ab133256, 1:3000). Immunoblots were imaged on a LI-COR ODYSSEY CLx imager and analyzed using Image Studio™ version 5.2.

**Reporter assay for astrovirus replicon activity**. BSR and Huh7.5.1 cells were transfected in triplicate with Lipofectamine 2000 reagent (Invitrogen), using the protocol in which suspended cells are added directly to the RNA complexes in 96-well plates. For each transfection, 100 ng replicon plus 10 ng firefly luciferase (FLuc)-encoding purified T7 RNA (RNA Clean and Concentrator, Zymo research) plus 0.3 μL Lipofectamine 2000 in 20 μL Opti-Mem (Gibco) supplemented with RNaseOUT (Invitrogen; diluted 1:1000 in Opti-Mem) were added to each well containing $10^5$ cells. Transfected cells in DMEM supplemented with 5% FBS were incubated at 37 °C for the indicated times (Fig. 4c, d, f). FLuc and RLuc activities were determined using the Dual Luciferase Stop & Glo Reporter Assay System (Promega). Replicon activity was calculated as the ratio of Renilla (subgenomic reporter) to Firefly (cap-dependent translation, loading control), normalized by the same ratio for the wt O2RL replicon sequence. Three independent experiments each in triplicate were performed to confirm reproducibility of the results.

**Virus inhibition assay**. The virus inhibition assay was performed in Caco2 cells in quadruplicate at MOI 0.2 in the presence or absence of 5 μM hexamethylene amiloride (HMA, Sigma, A9561). After infection for 48 h, media was completely removed and tested for virus- and drug-induced cellular toxicity using the CyQUANT LDH cytotoxicity assay (Thermo Scientific). The remaining cells were resuspended in HMA-free serum-free media and, after three freeze-thaw cycles, resulting viral stocks were titrated on Caco2 cells.

**Statistics and reproducibility**. Experiments in Fig. 5a, b, c, d, h and Supplementary Figs. 13f, 15 and 19d were repeated independently three times with similar results. *P* values in Fig. 6c, i come from two-tailed *t*-tests with separate variances and without adjustment for multiple comparisons.

**Reporting summary**. Further information on research design is available in the Nature Research Reporting Summary linked to this article.

## Data availability

The sequencing data reported in this paper (Fig. 2 and Supplementary Fig. 10) have been deposited in ArrayExpress (http://www.ebi.ac.uk/arrayexpress) under the accession number E-MTAB-8045. Virus sequence data for comparative genomic analyses were obtained from the National Center for Biotechnology Information GenBank nucleotide database (https://www.ncbi.nlm.nih.gov/nucleotide/). Source data are provided with this paper.

## Code availability

The SYNPLOT2, Ribo-Seq data analysis pipeline, and astrovirus sequence processing code are freely available in the GitHub repositories https://github.com/AndrewFirth12/synplot2, RiboseqAnalysis and AstrovirusORFx, respectively. Source data are provided with this paper.

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

## Acknowledgements

We thank the Cambridge NIHR BRC Cell Phenotyping Hub for access to confocal microscopy and Cambridge Genomic Services for the high-throughput sequencing. We thank Susanne Bell for technical assistance, Vanesa Madan, Alfredo Castello, Jia Lu, Ian Goodfellow, Myra Hosmillo, and Colin Crump for valuable advice, Susana Guix for providing the HAstV4 isolate, Betty Chung for reagents, Polly Roy for BSR cells, and Ian Brierley and Vanesa Madan for critical reading of the paper. The pAVIC1 infectious clone originally developed by Suzanne Matsui was provided by Stacey Schultz-Cherry. This work was supported by Wellcome Trust grant [106207] and European Research Council grant [646891] to A.E.F.

## Author contributions

A.E.F. and V.L. conceived the project. V.L. performed the experiments. A.E.F. performed the bioinformatic analyses. V.L. and A.E.F. wrote the paper.

## Competing Interests

The authors declare no competing interests.
