## [Peer Review File · Nature Communications]

Reviewers' Comments:

Reviewer #2:

Remarks to the Author:

In this revised manuscript, the authors provide compelling new data to support their conclusion that astroviruses encode a unique gene which can function as a viroporin. The manuscript is well-written and will be of great interest to the virology community. No further comments.

Reviewer #4:

Remarks to the Author:

The manuscript revisions have addressed my concerns with the original submission. The statistical analysis of the ribosome profiling data clearly supports the inference of ORFX translation, and the new data on the membrane association, topology, and multimerization of ORFX are clear.

Reviewer #5:

Remarks to the Author:

The authors claim to have found, via comparative genomic analysis of hundreds of astrovirus sequences, a novel viroporin. A conserved ORF (ORFX) overlaps the 5' region of ORF2 in some genogroups (I, III and IV). Clades with synonymous site conservation in that region also contain a conserved overlapping +1 frame ORF. This extra ORF was predicted previously in group I by the authors, and encoded a 112 aa peptide, and other sequences described in the present paper have similar lengths, all in the range expected for viroporins.

Initiation of this ORF is confirmed by ribosome profiling of infected cells, and whether XP is important is shown by using XP-knockout astroviruses, or XP severe truncations. The escape mutants observed recover start codon and remove introduced stop codons in XP.

The main concern of reviewer 1 (also shared by the other reviewers) is insufficient evidence that XP is a viroporin. In this updated version, the authors show that XP monomers form homo-oligomers (at least homo-dimers) and that these are disrupted by mutations at the predicted TM domain (Fig. 5h), therefore the C-terminal domain seems to be involved in this oligomerization. Further evidence of membrane insertion, and not mere association, is provided by the differential detection of mCherry, in N- or C-terminal tagged XP.

The authors also show increased permeability to hygromycin and that this is clearly related to XP by the use of several mutants.

Finally they show that HMA exposure has an effect on viral inhibition in presence, but not in absence, of functional XP.

One apparent problem is heterogeneity: some sequences do not have a predicted ORFX, others have no predicted hydrophobic stretches consistent with transmembrane-spanning motifs even though ORFX is predicted, and others have 1 or in some cases 2 predicted TMs (Figs S8 and S9). I would not waste much time in viroporin 'class' classifications. First, as a concept is not that useful since it is not related to function. Second, what appears to be 'predicted TMs' may not be, if secondary prediction is

not α -helical. It seems likely that evolution has found several solutions for the different genogroups and for some, specialization of XP has not been found necessary. Additionally, XP evolution should be constrained by the embedding of ORFX in ORF2, and the presence of one or multiple charged residues in the middle of the C-terminal putative TM may be a hint of this. To accommodate these constraints, the α -helix may not be canonical, having more or less residues per turn, or it may be kinked. I find these are logical arguments.

A complete definition of viroporins is a short peptide (typically ~50-150 a) that has at least one TM domain and displays ion channel activity driven by homo-oligomerization (typically tetramers to hexamers). This paper does not prove that, as neither oligomeric size > 2 or ion channel activity are shown. However, I find there is sufficient evidence that at least some of these sequences have a TM domain, form some kind of oligomer and permeabilize host membranes. These are all features of known viroporins and the paper presents compelling evidence for that.

However I disagree with the authors that 'we have certainly found a key function – and likely the main function – of XP'. XP may have critical functions related to host or viral protein-protein interaction or ion channel activity. Membrane permeabilization to a relative large molecule like hygromycin B is certainly not 'the role of XP', but just a consequence of its membrane-disrupting capabilities. Full characterization in the infected cell has taken decades for other viroporins, and at a minimum requires structural and channel activity characterization, which is clearly out of the scope of this paper. The constraints imposed by the location of this ORF may have resulted in unique structural solutions.

Response to reviews

We thank all the reviewers for their additional comments on the manuscript. We have carefully considered the remaining comments and have provided our responses below.

REVIEWERS' COMMENTS:

Reviewer #2 (Remarks to the Author):

In this revised manuscript, the authors provide compelling new data to support their conclusion that astroviruses encode a unique gene which can function as a viroporin. The manuscript is well-written and will be of great interest to the virology community. No further comments.

We thank the reviewer for their kind comments.

Reviewer #4 (Remarks to the Author):

The manuscript revisions have addressed my concerns with the original submission. The statistical analysis of the ribosome profiling data clearly supports the inference of ORFX translation, and the new data on the membrane association, topology, and multimerization of ORFX are clear.

We thank the reviewer for their kind comments.

Reviewer #5 (Remarks to the Author):

The authors claim to have found, via comparative genomic analysis of hundreds of astrovirus sequences, a novel viroporin. A conserved ORF (ORFX) overlaps the 5' region of ORF2 in some genogroups (I, III and IV). Clades with synonymous site conservation in that region also contain a conserved overlapping +1 frame ORF. This extra ORF was predicted previously in group I by the authors, and encoded a 112 aa peptide, and other sequences described in the present paper have similar lengths, all in the range expected for viroporins.

Initiation of this ORF is confirmed by ribosome profiling of infected cells, and whether XP is important is shown by using XP-knockout astroviruses, or XP severe truncations. The escape mutants observed recover start codon and remove introduced stop codons in XP.

The main concern of reviewer 1 (also shared by the other reviewers) is insufficient evidence that XP is a viroporin. In this updated version, the authors show that XP monomers form homo-oligomers (at least homo-dimers) and that these are disrupted by mutations at the predicted TM domain (Fig. 5h), therefore the C-terminal domain seems to be involved in this oligomerization. Further evidence of membrane insertion, and not mere association, is provided by the differential detection of mCherry, in N- or C-terminal tagged XP.

The authors also show increased permeability to hygromycin and that this is clearly related to XP by the use of several mutants.

Finally they show that HMA exposure has an effect on viral inhibition in presence, but not in absence, of functional XP.

One apparent problem is heterogeneity: some sequences do not have a predicted ORFX, others have no predicted hydrophobic stretches consistent with transmembrane-spanning motifs even though ORFX is predicted, and others have 1 or in some cases 2 predicted TMs (Figs S8 and S9). I would not waste much time in viroporin 'class' classifications. First, as a concept is not that useful

since it is not related to function. Second, what appears to be 'predicted TMs' may not be, if secondary prediction is not α -helical. ...

We have now deleted the only reference to viroporin 'class' classifications: ", thus making XP a candidate class IA viroporin, a class which also includes the influenza A virus M2, coronavirus E and HIV-1 Vpu proteins".

... It seems likely that evolution has found several solutions for the different genogroups and for some, specialization of XP has not been found necessary. Additionally, XP evolution should be constrained by the embedding of ORFX in ORF2, and the presence of one or multiple charged residues in the middle of the C-terminal putative TM may be a hint of this. To accommodate these constraints, the α -helix may not be canonical, having more or less residues per turn, or it may be kinked. I find these are logical arguments.

These comments are in support of our arguments presented in the manuscript and we agree with them.

A complete definition of viroporins is a short peptide (typically ~50-150 a) that has at least one TM domain and displays ion channel activity driven by homo-oligomerization (typically tetramers to hexamers). This paper does not prove that, as neither oligomeric size > 2 or ion channel activity are shown. However, I find there is sufficient evidence that at least some of these sequences have a TM domain, form some kind of oligomer and permeabilize host membranes. These are all features of known viroporins and the paper presents compelling evidence for that.

However I disagree with the authors that 'we have certainly found a key function – and likely the main function – of XP'. XP may have critical functions related to host or viral protein-protein interaction or ion channel activity. Membrane permeabilization to a relative large molecule like hygromycin B is certainly not 'the role of XP', but just a consequence of its membrane-disrupting capabilities.

Full characterization in the infected cell has taken decades for other viroporins, and at a minimum requires structural and channel activity characterization, which is clearly out of the scope of this paper. The constraints imposed by the location of this ORF may have resulted in unique structural solutions.

We thank the reviewer for their supportive comments.

Although the reviewer mentions a few concerns, our interpretation is that they are generally happy with the way in which we have dealt with them in the current version of the manuscript. Our interpretation is that the two items where the reviewer might like to see minor modifications is (1) To mention additional key features of viroporins which need to be addressed in the future (structure of the potential ion channel, its oligomeric state, and ion specificity), and (2) to discuss that XP may have additional important functions besides those so far discovered. The second point relates mainly to a comment ("key function – and likely the main function") in our previous rebuttal but not in the manuscript itself. Here we were referring less to XP's specific viroporin-like activity and more to XP's effect of promoting efficient virus assembly/release, as supported by our analysis to determine which stage of the virus life cycle is affected by XP knockout.

Addressing both points in the manuscript, we already have the text:

"Future work will be needed to confirm potential ion channel activity, and characterize ion specificity, structural organization, and any additional processes affected by XP expression in the context of viral infection. The localization of XP not only at the plasma membrane but also in the perinuclear TGN membranes raises the possibility that XP may also have additional functions. Whereas the C-terminal α -helix appears to be associated with the cell-permeabilizing activity, additional functions (if any) of the extended N-terminal domain of XP remain to be studied."

We have now also added "oligomeric state" to read: "Future work will be needed to confirm potential ion channel activity, and characterize ion specificity, oligomeric state and structural organization, ...".

Also, to avoid overinterpreting the hygromycin B permeabilization assay, we have also modified "All tested XPs resulted in increased cell permeabilization to HB (Fig. 6c) confirming conservation of XP activity." to "... confirming conservation of XP activity when tested with this assay."